# Tensile Properties of As-Built 18Ni300 Maraging Steel Produced by DED

Jorge Gil [1,2,*] , Ricardo Seca [1] , Rui Amaral [1,2] , Omid Emadinia [2] , Abílio De Jesus [1,2] and Ana Reis [1,2]

1   Faculdade de Engenharia, Universidade do Porto, s/n, R. Dr. Roberto Frias, 4200-465 Porto, Portugal
2   LAETA/INEGI: Instituto de Ciência e Inovação em Engenharia Mecânica e Engenharia Industrial, Campus da FEUP, R. Dr. Roberto Frias 400, 4200-465 Porto, Portugal
*   Correspondence: jgil@inegi.up.pt

**Abstract:** The mechanical behaviour of as-built DED-produced 18Ni300 Maraging steel was studied by manufacturing a wall-like structure from which three different specimen types were obtained: specimens in which the loading direction was the same as the printing direction (vertical), specimens in which these two directions were perpendicular (horizontal), and bimetallic specimens in which the interface between the AISI 1045 substrate and the 18Ni300 steel was tested. The yield strength of the produced samples was $987.9 \pm 34.2$, $925.9 \pm 89.7$ and $486.7 \pm 47.2$ MPa for the vertical, horizontal and bimetallic specimens, respectively, while the elongation to failure was $9.4 \pm 1.9$, $18.3 \pm 2.3$ and $14.06 \pm 0.6\%$ in the same order. The latter specimen failed within the substrate-comprised portion of the specimen. Additionally, the fracture surfaces were analysed through scanning electron microscopy, concluding that while both surfaces consist of dimples, the horizontal specimen presented microporosities with a reduced diameter. A microhardness analysis in the printed wall-like structure following the printing direction yielded an average hardness of $392 \pm 21$ HV0.3, with fluctuations along the build direction mostly within one standard deviation.

**Keywords:** additive manufacturing; directed energy deposition; maraging steel; tensile tests; digital image correlation; microstructural analysis

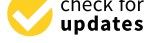



## 1. Introduction

Additive manufacturing (AM) is a set of technologies in which three-dimensional components are built on a layer-by-layer basis. One of the possible classifications of metallic AM processes distinguishes powder-bed and non-powder-bed techniques, with directed energy deposition (DED) belonging to the latter. One of the advantages of not requiring the spread of a fresh powder layer at each height increment is the ability to locally repair already existing components, which constitutes one of the most attractive advantages of DED processes when compared to selective laser melting (SLM) or selective laser sintering (SLS). In DED, the metallic feedstock is fed directly into the path of the heat source (which may be a laser or an electron beam) through a powder splitter and nozzle head, both mounted onto a numerically controlled system. The system also relies on an inert gas supply to shield the melting pool of corrosion phenomenon. Although DED processes have better productivity than SLM, the surface quality and geometric fidelity is usually worse, thus requiring subtractive operations such as machining before the component is ready for its intended application.

DED techniques have been used to successfully produce claddings and/or fully three-dimensional components with different alloys, such as titanium [1,2], nickel [3,4], copper [5] and aluminium [6] alloys, tool steels [7,8] and stainless steels [9,10]. One additional material group that has been DED-processed are Maraging steels, which consist of high-strength, precipitation hardened steels. This research work focuses on 18Ni300, which contains 18% Ni, substantial presence of Co and Mo, as the former promotes age hardening by reducing

the solubility of Mo in the matrix [11], while the latter forms the intermetallic compounds that precipitate, enhancing the alloys' strength [12]. Additionally, small quantities of Ti and Al (0.05–0.5% and 0.05–0.15%, respectively), further contribute to intermetallic phases to be precipitated [12]. These alloys find their applications in military [13,14], aircraft [15], aerospace [16,17] and die-casting and tooling industries [18,19].

Recent research has been focusing on Maraging processing through DED: Félix-Martínez et al. [20] optimised the process parameters of 18Ni300 and built processing maps according to five different objectives, with these being dilution proportion, height, width, depth and porosity. J. Gil et al. [21] optimised the processing parameters of 18Ni300 on AISI H13, reaching a set of parameters of a laser power of 1850 W, a scanning speed of $12 \, \mathrm{mm \, s^{-1}}$, feeding rate of 12 g/min, shielding gas flow rate of 25 L/min and carrier gas flow rate of 4 L/min. K. Aleksandr et al. [22] studied the effect of deposition parameters on single-bead depositions, as well as the residual stresses on a bulk cube, concluding that an energy density of $34.7 \, \mathrm{J \, mm^{-2}}$ yielded a fully dense specimen with relatively low residual stresses. L. Zhang et al. [23] wrote a review article which explores the use of Maraging alloys containing 18% Ni content within metallic additive manufacturing processes. Bo Chen et al. [24] studied the influence of processing parameters of DED-processed 18Ni300 by testing tensile specimens produced through nine different conditions following an orthogonal Taguchi array, concluding that the optimised set reached a ultimate tensile strength of 959.2 MPa. M. Polański et al. [25] reported the influence of layer thickness on mechanical properties of heat-treated samples, obtaining ultimate tensile strengths of 1958 MPa in the samples with the lowest tested layer thickness of 0.5 mm. S. Amirabdollahian et al. [26] analysed the thermal cycles' influence on previously deposited layers in an 18Ni300 by varying the inter-layer idle time, thus inducing an in situ aging effect, successfully triggering nano-precipitation of intermetallics that enhanced the sample's hardness and strength. Regarding other martensitic precipitation-hardenable steels, R. Mendagaliev et al. [27] studied different heat treatments in DED-processed CA6NM, concluding that a repeated 2 h cycle at 640 °C led to a set of mechanical properties closest to its cast counterpart. G. Wang [28] studied the cooling rate influence in phase transformation and interface bonding strength of FV520B produced through wire DED by depositing a clad surface over a rod and subjecting the sample to a tensile quasi-static load, reaching 1046 MPa at a rate of $0.05 \, \mathrm{mm \, s^{-1}}$.

As aforementioned, one of the prominent applications of DED technologies is component repair. Extending a component's life cycle is attractive from both an economic perspective, as large components such as casting dies are expensive to produce [29], but also from an environmental perspective, as it leads to less material waste. Recent research focused on using DED as a repairing tool includes the study of deposition strategy and fill depth in component properties by C. Barr et al. [30], which studied the monotonic and fatigue behaviour of several specimens with distinct percentages of added 18Ni300 material, analysing the intergranular cracking as the main crack initiation mechanism. J. Bennet et al. [31] studied the mechanical reliability and enviornmental impact of repairing an automotive die through DED using H13, inferring a better performance than conventional TIG welding repair as the DED-remanufactured die achieved the same life as the original die. S. Skerlos et al. [32] analysed the environmental impact of laser-based DED refurbishment of dies, reaching the conclusion that the larger the solid-to-cavity ratio, the more environmentally costly it becomes to repair through DED technologies.

Maraging steels' use within die repairing via DED is justified by its mechanical strength, acceptable corrosion resistance [30] and good weldability [23]. Nevertheless, it is necessary to understand its properties after DED processing, as well as evaluate the adhesion between printed geometries and the substrate. In order to further understand and validate the use of Maraging in die repair, an 18Ni300 Maraging steel was deposited on an AISI 1045 steel plate, and subsequently electrically discharge machined to obtain three specimen types, enabling the study of 18Ni300's properties in two distinct directions

in relation to the building direction, as well as a bimetallic specimen comprised of the deposited 18Ni300 and the AISI 1045 baseplate.

## 2. Materials and Methods

### 2.1. Powder and Substrate

The powder was the W722 Böhler Edelstahl (Kapfenberg, Austria), an equivalent to the 18Ni300 grade, produced by inert gas atomisation [33]. Its morphology was assessed through scanning electron microscopy (SEM) using a FEI Quanta 400 FEG (FEI Company, Hillsboro, OR, USA), while the powder size distribution was ascertained via dynamic light scattering (DLS) in a Coulter LS399 (Beckmen Coulter, Brea, CA, USA) by suspending a sample of powder in ethanol and subjecting the solution to ultrasonic vibrations during one minute.

Figure 1 shows the morphology of the used powder, highlighting a large number of lumps around larger particles. The reason for this is unknown. The particle size distribution (PSD) is given in Figure 2. The $D_{10}$, $D_{50}$ and $D_{90}$ are 48.8 μm, 89.8 μm and 143.7 μm, respectively, in line with the nominal values given by Böhler [33].

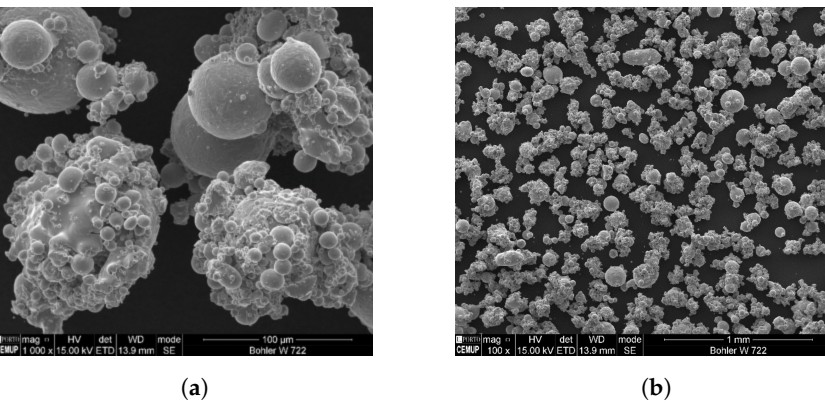

(**a**)          (**b**)

**Figure 1.** Powder morphology, as obtained through scanning electron microscopy: (**a**) highlights the powder particle agglomerates that characterise the used powder; (**b**) shows a general configuration of the analysed powder sample.

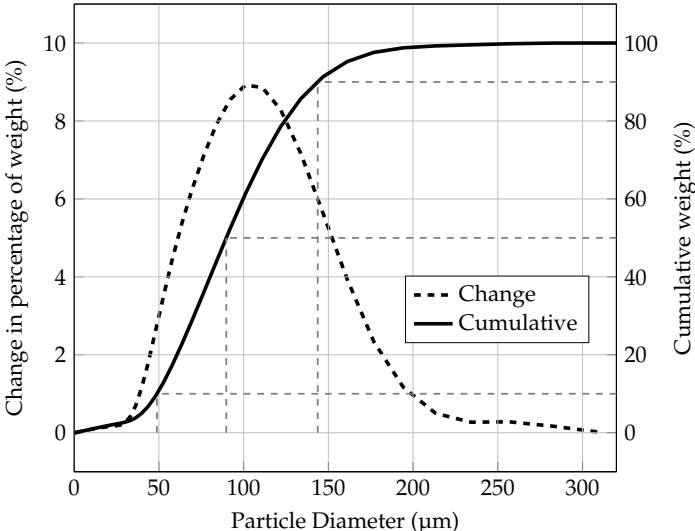

**Figure 2.** W722 particle size distribution obtained through dynamic light scattering.

Regarding the substrate, F10 steel (FRamada, Ovar, Portugal) was used, being an equivalent grade to AISI 1045 (DIN CK45) steel, supplied in its non-heat-treated state [34]. A chemical analysis was conducted through optical emission spectroscopy (OES) and

compared to the nominal values provided by the supplier, as shown in Table 1. The supplied plates had 25 mm thickness and were subjected to grinding for reduced surface roughness.

**Table 1.** Nominal (N) and measured (M) chemical composition of AISI 1045, used as the substrate in the performed depositions.

| Material | | Chemical Composition [%] | | | | | | | | |
|---|---|---|---|---|---|---|---|---|---|---|
| | | Cr | Mo | Mn | P | C | Si | V | S | Fe |
| AISI 1045 | N | - | - | 0.5–0.8 | ≤0.035 | 0.42–0.5 | ≤0.40 | - | - | Bal. |
| | M | 0.02 | 0.01 | 0.66 | ≤0.001 | 0.46 | 0.17 | - | ≤0.003 | Bal. |

### 2.2. DED Setup

The directed energy deposition setup used throughout the present work consisted of a Coherent Highlight FL3000 fibre laser (Coherent, Santa Clara, CA, USA), with the capacity of continuous wave operation with power up to 3000 W and a laser spot size of 2.5 mm, two medicoat AG disk powder feeders (Medicoat, Mägenwill, Switzerland), a Kuka 6 axis robot (KUKA, Augsburg, Germany), in which a COAX12V6 nozzle (Fraunhoffer IWS, Dresden, Germany), a Fraunhoffer IWS powder splitter and a Fraunhoffer E-MAqS system were mounted. Gas supply for corrosion shielding and powder carrying was regulated at 6 bar, and Argon with purity ≥ 99.99% from Nippon Gases (Maia, Portugal) was used. Figure 3 shows several of the components.

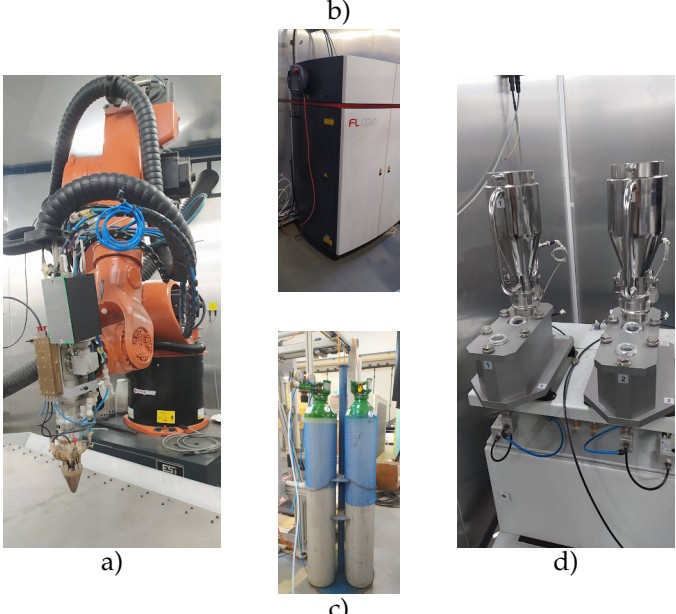

**Figure 3.** DLD system used for the performed depositions: (**a**) shows the Kuka 6-axis robotic arm mounted with the powder nozzle, powder splitter, and E-MAqS temperature control system; (**b**) shows the fibre laser; (**c**) the argon cannisters that fuel the shielding and powder carrying gas, while (**d**) shows the disk powder feeders containing the used feedstock.

### 2.3. Specimen Printing Process

The deposition from which specimens were machined consisted on a $110 \times 6 \times 45$ mm wall component, whose process parameters were the ones indicated in Table 2. The parameter choice was based on a previously undergone process parameter optimisation [35], as well as the existing literature on 18Ni300 DED-processing. The process parameters are shown in Table 2. The resulting energy density was defined by Equation (1)

$$E_d = \frac{P}{v_s \times d_s} \tag{1}$$

where $E_d$ is the energy density, $P$ is the laser power, $v_s$ is the scanning speed, and $d_s$ is the laser spot size diameter, which was $61.67\,\mathrm{J\,mm^{-2}}$.

**Table 2.** Process parameters used in the deposition of the deposited wall from which specimens were obtained. $P$ is the laser power, $v_s$ is the scanning speed, $f_r$ is the powder feeding rate, $q_s$ is the shielding gas flow rate, $q_r$ is the powder carrying gas flow rate, and $t_r$ is the laser spot size diameter.

| $P$ W | $v_s$ [mm s$^{-1}$] | $f_r$ [g min$^{-1}$] | $q_s$ [L min$^{-1}$] | $q_r$ [L min$^{-1}$] | $t_r$ [mm] |
|---|---|---|---|---|---|
| 1850 | 12 | 12 | 25 | 4 | 2.5 |

Three distinct types of specimens were obtained from the printed component: specimens in which the printing build direction and the loading direction were collinear (which are henceforth named vertical), specimens in which the aforementioned directions were perpendicular (henceforth named horizontal) and bimetallic specimens, whose loading volume comprised two distinct materials-the deposited 18Ni300 and the AISI 1045 substrate and the resulting interface. A schematic representation of these three specimens types and the configuration within the deposited walled geometry is shown in Figure 4a, while the specimen geometry is shown in Figure 4b. The manufacturing process through which the tensile specimens were obtained was wire electrical discharge machining (wEDM). Three specimens for each loading direction were machined. Their thickness, width and section area at the gauge length is shown in Table 3.

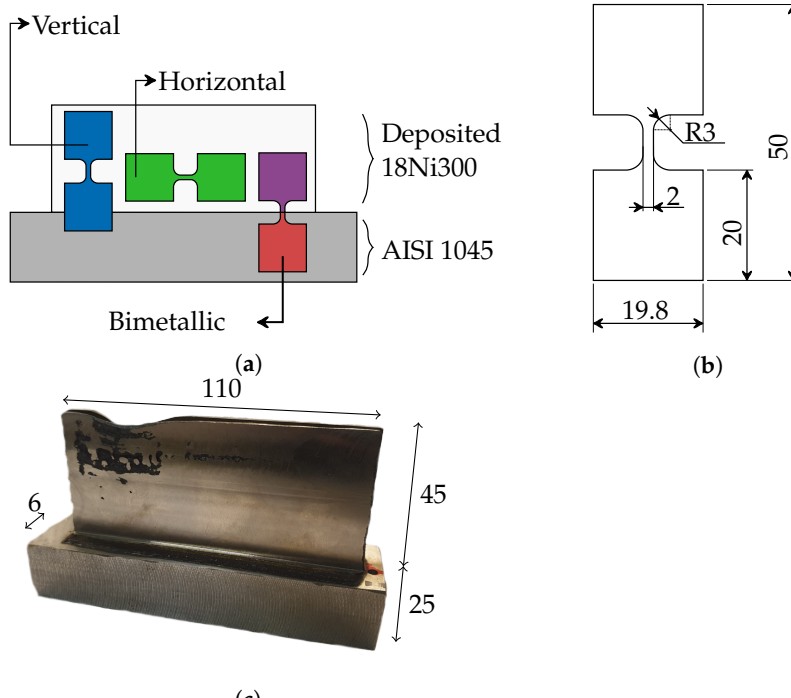

**Figure 4.** Tensile specimen manufacturing overview: (**a**) specimen configuration within printed geometry; (**b**) its geometry in millimetres, with a thickness of 2 mm; (**c**) the deposited structure on top of the AISI 1045 baseplate, with its dimensions annotated in millimetres.

**Table 3.** Specimen dimensions, measured at the gauge length. Three measurements were conducted along gauge length, with presented value being an average of the three values. Section area computed with average thickness and width values.

| Location | Specimen | Thickness [mm] | Width [mm] | Section Area [mm²] |
|---|---|---|---|---|
| Horizontal | MH1 | 1.031 | 2.079 | 2.14 |
| | MH2 | 1.034 | 2.088 | 2.16 |
| | MH3 | 0.964 | 2.074 | 1.99 |
| Vertical | MV1 | 1.038 | 2.090 | 2.17 |
| | MV2 | 1.024 | 2.084 | 2.13 |
| | MV3 | 1.019 | 2.084 | 2.12 |
| Bimetallic | MI1 | 1.033 | 2.083 | 2.15 |
| | MI2 | 1.038 | 2.088 | 2.17 |
| | MI3 | 1.026 | 2.075 | 2.13 |

### 2.4. Tensile Setup

The tensile tests were performed in an Instron 5900R machine with displacement control of $0.5\,\mathrm{mm\,min^{-1}}$, with displacement field measurements carried out by digital image correlation (DIC) at an acquisition rate of 10 Hz. As the testing area was uniform, an average speckle size of less than 15 µm was applied. The camera that captured the tests was a 5 Megapixel Baslerac (Basler AG, Ahrensburg, Germany) A2440-75 µm, $2448 \times 2048$ px with a telecentric InfaimonOPE-TC-23-9.45 mm lens (Infaimon, Aveiro, Portugal). The software for data post-processing, such as strain field computation and virtual extensometre, was VIC-2D.v6 (Correlated Solutions, Kassel, Germany).

## 3. Results and Discussion

### 3.1. 18Ni300 Mechanical Characterisation

The results of the performed tensile tests are shown in Figure 5a,b for the engineering stress-engineering strain curves true stress–true strain, respectively.

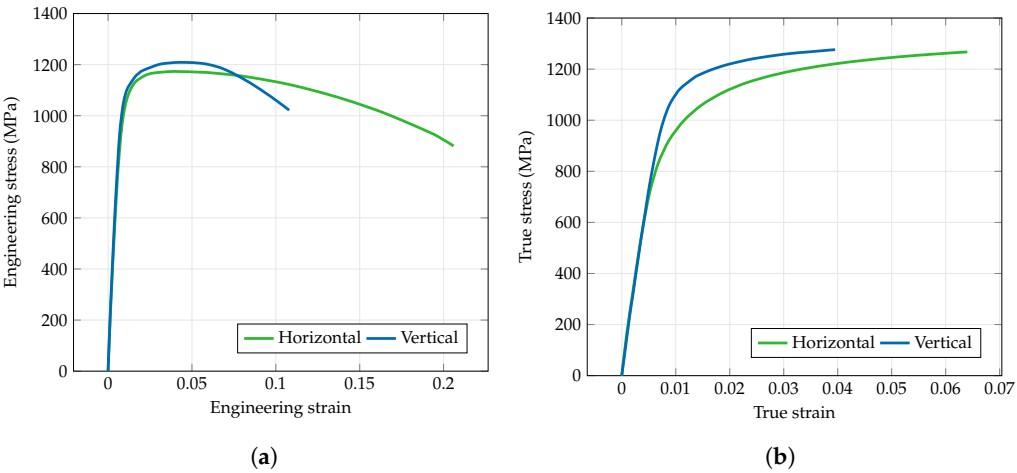

(**a**)            (**b**)

**Figure 5.** Monotonic tensile test results for the deposited 18Ni300 specimens: engineering stress-engineering-strain curves (**a**) and true stress–true strain (**b**) curves.

The yield strength, ultimate tensile strength and elongation to fracture of deposited 18Ni300 are shown in Table 4. The mechanical strength between the horizontal and vertical specimens is relatively similar, with the average yield strength and ultimate tensile strength in both directions varying by 6.27% and 1.71%, respectively. This similarity in material strength between differently orientated specimens is expected, considering that despite the difference in cutting procedure, both specimens are comprised of the same material. The difference in metallurgical features amongst the different specimens is predominately observed in the difference of plastic behaviour, as explained further in this work. Research on the mechanical

properties of differently oriented DED-produced samples reached similar conclusions, with the yield strength and ultimate tensile strength of DED-W deposited Ti6Al4V by J. Alexis et al. [36] in the horizontal and vertical directions showing a difference of 2.59 and 6.34% for the yield strength and ultimate tensile strength, respectively. C. Saldana et al. [37] concluded that the build orientation effect on mechanical properties of DED-produced 316 did not have statistical significance, according to the ANOVA method, in the samples' ultimate tensile strength. The disparity in the results within the horizontal direction is considerable, with the yield strength oscillating between 810.5 and 1029.1 MPa. The difference in the total elongation between the horizontal and vertical specimens was also of note, with the horizontal specimen displaying more ductility on average than its vertical counterpart. This is partly expected due to the nature of additive processes, as newly deposited layers require the remelting of previously solidified material, generating microstructural anisotropy that affects the properties of the component, as reported in similar research [38,39]. Other research reports the presence of weak interfacial layers as a cause for the recorded anisotropy [40].

**Table 4.** Obtained mechanical properties of the deposited 18Ni300, according to two different specimen extraction configurations.

| Specimen | Yield Strength [MPa] | Ultimate Tensile Strength [MPa] | Total Elongation [%] |
|---|---|---|---|
| Horizontal | $925.9 \pm 89.7$ | $1188.6 \pm 12.0$ | $18.3 \pm 2.3$ |
| Vertical | $987.9 \pm 34.2$ | $1209.4 \pm 16.5$ | $9.4 \pm 1.9$ |

A comparison regarding the ultimate tensile strength $\sigma_{uts}$ in the literature for DED-processed 18Ni300 in its as-built condition was performed, with the results shown in Figure 6. The ultimate strength's larger values for the AM-produced parts is not uncommon [23,40], considering the fine microstructure that arises from the fast cooling rates associated with AM [23]. As observable through Figure 6b, there is no immediate conclusion regarding the relation between the material's strength and energy density during deposition; B. Chen (2018) indicated that the most significant parameter in the resulting mechanical properties was the scanning speed [24], followed by the laser power and powder feeding rate. The used scanning speed in B. Chen's 2018 work was $15\,\mathrm{mm\,s^{-1}}$, while S. Amirabdollahian's sample was produced using $21.67\,\mathrm{mm\,s^{-1}}$.

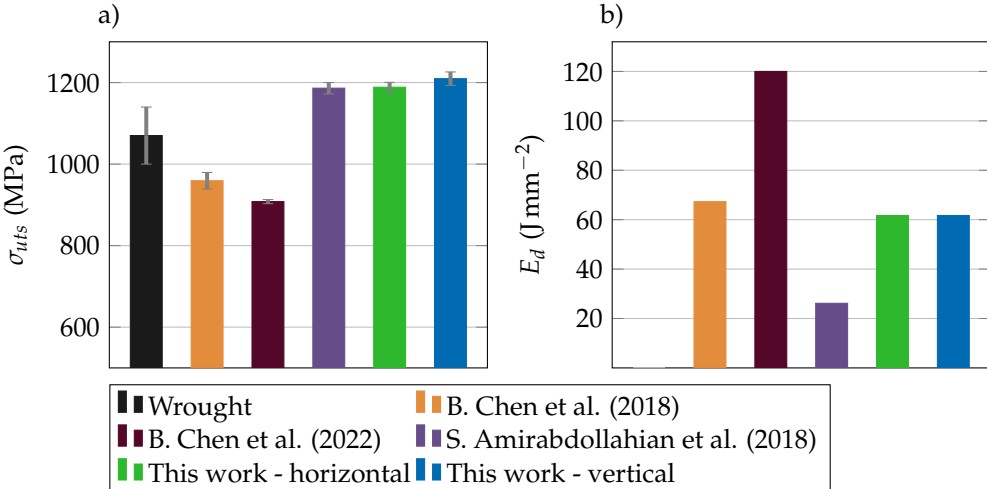

**Figure 6.** Visual representation of the comparison between the as-built ultimate tensile strength $\sigma_{uts}$ of DED-produced 18Ni300 (**a**) and the energy density $E_d$ value that produced the resulting specimens (**b**). Wrought material values were adapted from reference [41] (1990, ASM International), from material in its non-heat-treated state. Remaining values were adapted from the following references: B. Chen (2018) et al. [24] (2018, Elsevier Ltd.), B. Chen et al. (2022) [42] (2022, Elsevier Ltd.), S. Amirabdollahian et al. [26] (2021, Acta Materialia Inc.). B. Chen et al. (2022)'s result is influenced by the sample's exposure to oxygen by a fraction of 50 ppm.

### 3.2. Bimetallic Specimen Characterisation

The results of the bimetallic specimen containing both AISI 1045 and deposited 18Ni300 are shown in Figure 7 and Table 5. The DIC-computed strain field in the loading direction of the specimen during imminent rupture is shown in Figure 8, highlighting the necking phenomena that occurred exclusively in the region comprised of AISI 1045, indicating good adhesion between the substrate and deposited material. Assuming good adhesion between materials, it is expected that the specimen fractures in the AISI 1045 zone, considering its lower yield strength and ultimate tensile strength, which the literature suggests to be 310 MPa and 565 MPa [43], both considerably smaller than 18Ni300's obtained yield strength of 925 MPa, further implying that the latter material only experienced elastic strain during the test.

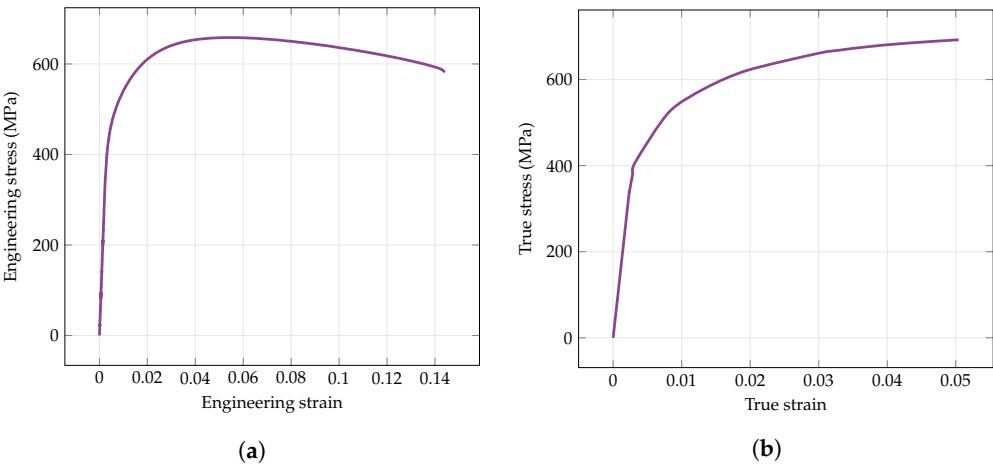

(**a**)  (**b**)

**Figure 7.** Monotonic tensile test results for the deposited bimetallic specimens: engineering stress-engineering strain (**a**) and true stress–true strain (**b**) curves.

**Table 5.** Obtained mechanical properties of the bimetallic specimens, comprised of the substrate (AISI 1045) and deposited alloy (18Ni300).

| Specimen | Yield Strength [MPa] | Ultimate Tensile Strength [MPa] | Total Elongation [%] |
|---|---|---|---|
| Bimetallic | $486.7 \pm 47.2$ | $663.4 \pm 4.6$ | $14.6 \pm 0.6$ |

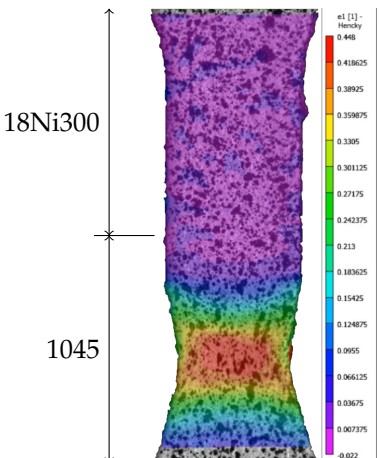

**Figure 8.** DIC-computed strain field along the loading direction of the bimetallic specimen.

### 3.3. Fracture Analysis

Microscopic observations performed by SEM on the fractured tensile specimens revealed the different behaviours that horizontal and vertical specimens showed. As shown

in Figures 9 and 10, the horizontal specimen suffered more plastic deformation, Figure 9a,c, than that of the vertical one, Figure 10a,c, i.e., necking effect occurred in the horizontal specimen. The fracture surfaces of both specimens consist of dimples, Figures 9b and 10b, though the size of microporosities in the vertical specimen is much larger than that of the horizontal one. This difference led to a premature fracture occurring in the vertical specimen, having smaller plastic deformation, influenced by the concentration of porosities at the interface of consecutive layers, Figure 10d, that acted as stress concentration zones.

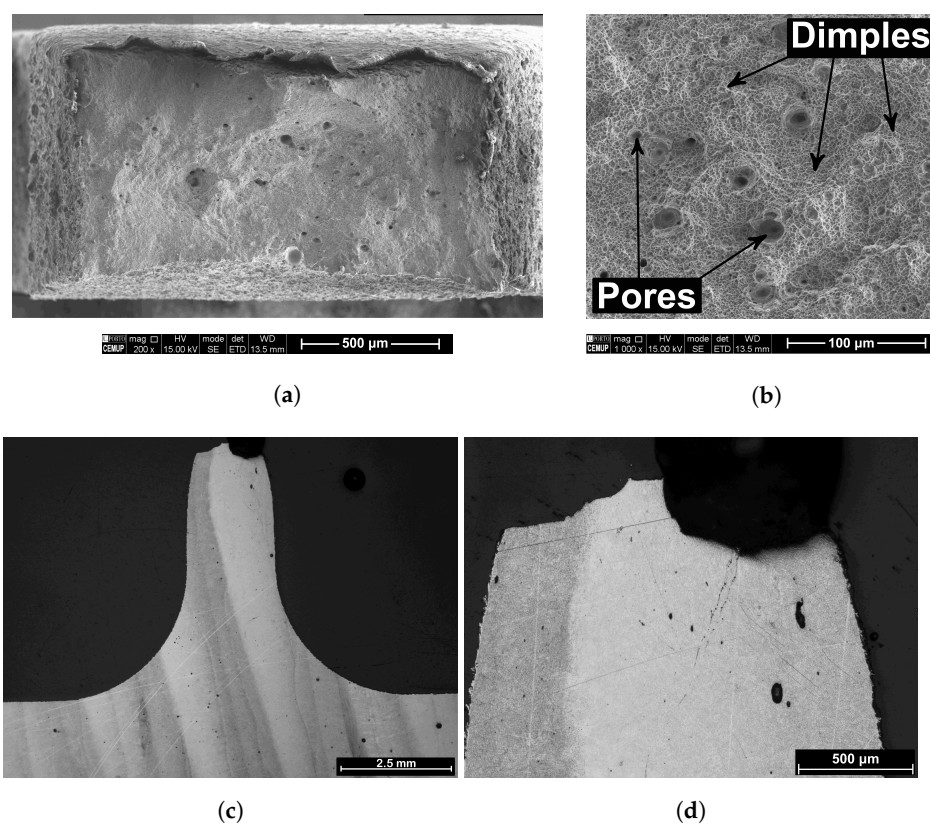

(a)

(b)

(c)

(d)

**Figure 9.** Horizontal specimen: SEM images of the: (**a**) fracture surface; (**b**) dimples and microporosities observed; (**c**,**d**) optical microscopy images of the lateral side, etched surfaces.

### 3.4. Mechanical Properties Affected by Microstructure

The graph illustrated in Figure 11 presents the hardness distribution along the cross section of the printed wall from the substrate to the topmost region, a hardness of $392 \pm 21$ HV0.3. The presence of smaller hardness values in the bottom regions of the wall, though having less and smaller porosities, can be attributed to a partial annealing happening in those zones during the wall printing process. The heat released by the molten material can act as a heat source affecting the hardness of the material beneath. A similar effect was distinguished in a previous study revealing that the heat generated during consecutive layers changed the microstructure and the hardness of the heat affected zone [35]. The porosity was quantified through optical images shown in Figure 12a,b with the use of the ImageJ software, yielding results of 99.26% and 99.71% for the bottom and middle sample, respectively. This is consistent with the hardness evolution along the building direction, as there is no significant increase or decrease in microhardness values above or below the standard deviation. It is important to mention that the obtained porosities are consistent with components manufactured through DED [40].

Several researchers have sought to establish relationships between steel's hardness and its true yield or true tensile strength [44–47]. J.R. Cahoon et al. [47] developed a relationship between a steel's hardness and ultimate tensile strength, as shown in Equation (2),

$$\sigma_{uts} = \frac{H}{2.9} \left( \frac{n}{0.217} \right)^n \tag{2}$$

in which $H$ is the hardness expressed in $N\,mm^{-2}$, and $n$ is the strain hardening exponent. The hardening exponent $n$ for both horizontal and vertical directions was computed by curve fitting the strain hardening equation proposed by J. Hollomon [48] to the plastic portion of the tensile curve of the test specimens, for both directions. The applied curve-fitting method consisted of the least squares method. Hollomon's equation is shown in Equation (3),

$$\sigma = K\varepsilon^n \tag{3}$$

in which the $K$ is the strength coefficient. The computed values of $n$ for the horizontal and vertical direction yields 0.1365 and 0.1108, respectively. The ultimate tensile strength predicted by Equation (2) thus results in 1244 and 1230 MPa for the horizontal and vertical specimens, respectively. There is, therefore, strong agreement between the predicted ultimate tensile strength of the deposited structure and the tested specimens, following Cahoon's expression, as the average true tensile strength of the specimens are 1279 and 1261 MPa, resulting in a deviation of 2.74 and 2.43% for the horizontal and vertical specimens, respectively, in relation to the predicted values.

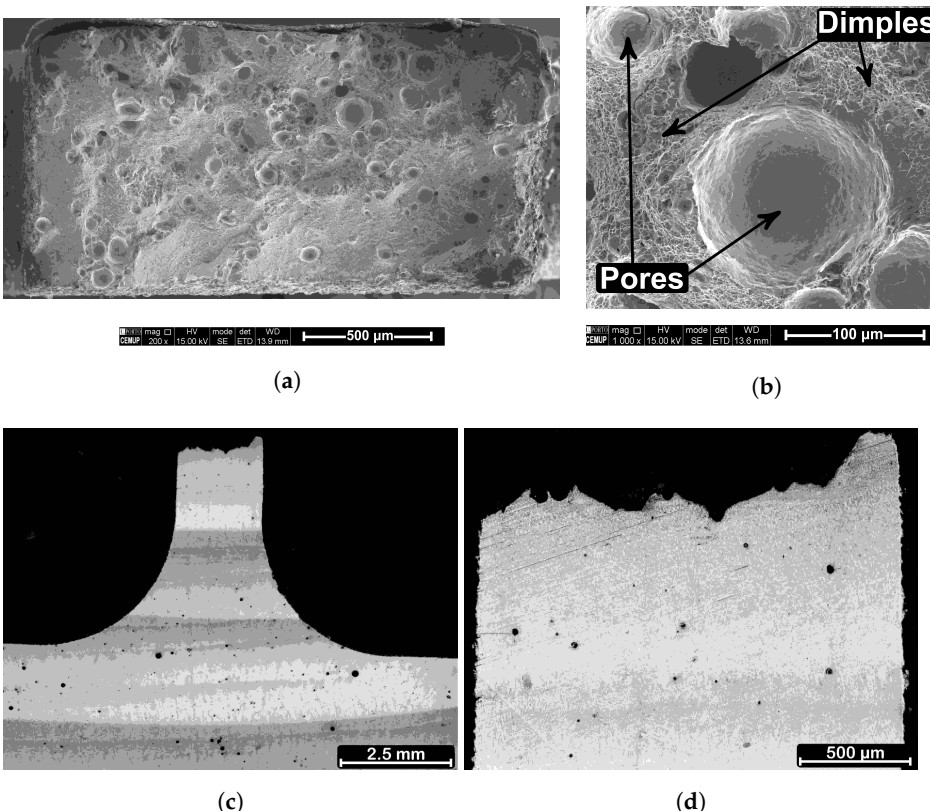

**Figure 10.** Vertical specimen: SEM images of the: (**a**) fracture surface (**b**) dimples and microporosities observed; (**c**,**d**) optical microscopy images of the lateral side, etched surfaces.

Regarding the deviation observed in the yield strength of the horizontal specimen, Table 4, this wide dispersion can be influenced by the distribution of microstructure, e.g., dendritic/grain size affected by the remelting/reheating of the consecutive printed lines/layers as shown in Figure 12c. Moreover, as seen in Figure 12a,b, the effective

gauge length of the tensile specimen in the vertical specimen seems to be constituted by a more homogeneous microstructure, excluding microporosities, than that of the horizontal specimen.

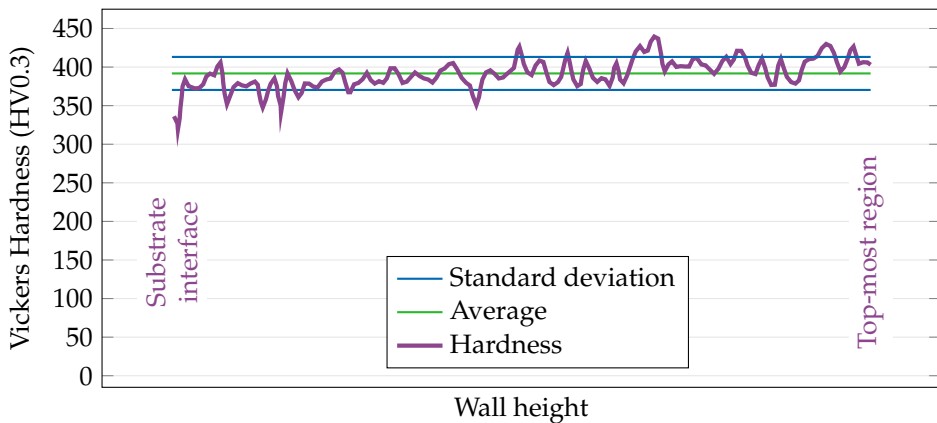

**Figure 11.** Microhardness graph across the printed wall from the bottom to the topmost region.

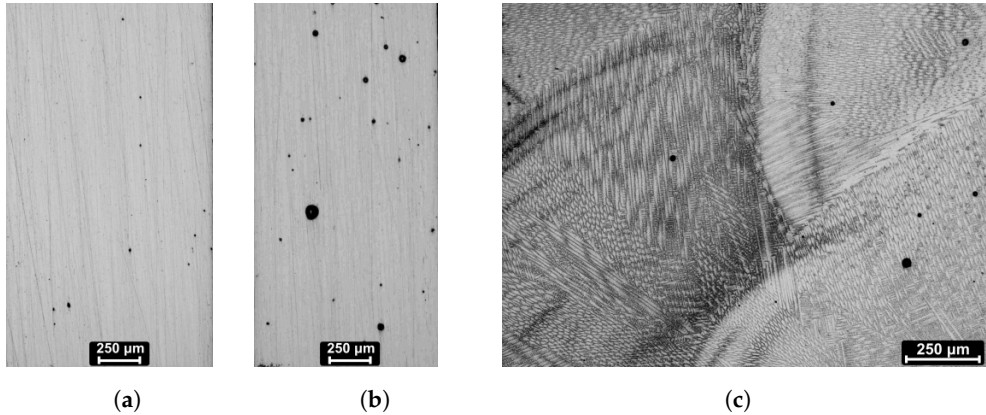

(**a**)          (**b**)          (**c**)

**Figure 12.** Optical microscopy images: (**a**,**b**) near the bottom and close to the middle of the printed wall; and (**c**) distribution of dendritic structure in different orientation observed in some neighbouring fused regions.

## 4. Conclusions

Three distinct samples of 18Ni300 Maraging steel were successfully manufactured, with the samples subsequently subjected to tensile testing, and the originated fracture surfaces analysed via SEM, as well as a microhardness analysis. The conclusions of these studies are as follows:

- The yield and ultimate tensile strengths of the horizontal and vertical specimens are similar, with the mean of each property varying 6.27% and 1.71%, respectively, between both samples, with larger values for vertical specimens;
- There is a significant dispersion in the stress–strain curves of the horizontal specimen, ranging from 810.5 to 1029.1 MPa, which may be attributed to the microstructural hetereogeneity resulting from the dendritic and grain size, influenced by remelting and reheating of successive layers;
- The mean of the elongation to failure in the horizontal specimen was 94.7% larger than the vertical specimen (18.3 and 9.4% in the horizontal and vertical, respectively), attributable to larger porosities in the latter specimen, which led to premature failure;
- The bimetallic specimen yielded at stresses in the range of AISI 1045's yield strength. The DIC-computed strain field instantly before fracture indicated that necking and subsequent fracture occurred entirely in the area comprised of AISI 1045, indicating good adhesion between the substrate and the deposited geometry;

- The fracture surface of the specimens was comprised of dimples, although the microporosity size in the vertical specimen was seen to be larger, indicating that the stress concentration in these specimens was larger, leading to premature failure.
- The average microhardness across the deposited wall-like geometry was $392 \pm 21$ HV0.3. The fluctuations along the geometry are attributed to the microporosity distribution.

**Author Contributions:** Conceptualization, O.E., J.G., R.A. and R.S.; methodology, O.E., R.S. and J.G.; validation, J.G. and O.E.; formal analysis, A.R. and A.D.J.; investigation, J.G. and R.S.; resources, A.R. and A.D.J.; writing—original draft preparation, J.G. and R.S.; writing—review and editing, A.R., A.D.J. and O.E.; visualization, J.G., R.A. and O.E.; supervision, A.R. and A.D.J.; funding acquisition, A.R. and A.D.J. All authors have read and agreed to the published version of the manuscript.

**Funding:** The authors would like to acknowledge the funding of the PhD scholarship with the reference UI/BD/150684/2021, funded by Fundação para a Ciência e Tecnologia (FCT). The authors acknowledge the funding through the Add.Strength project, with reference PTDC/EME-EME/31307/2017, funded by FEDER and FCT. The authors would further acknowledge the funding through the ADDing project, with reference POCI-01-0145-FEDER-030490, funded by FCT and FEDER. The authors would like to acknowledge the funding acquired through the MAMTool project, with reference PTDC/EME-EME/31895/2017, funded by FEDER and FCT. The authors would like to thank the support provided by Fundação para a Ciência e a Tecnologia of Portugal (FCT) and IDMEC under LAETA-UIDB/50022/2020.

**Conflicts of Interest:** The authors declare no conflict of interest.

## Abbreviations

The following abbreviations are used in this manuscript:

| | |
|---|---|
| AM | Additive manufacturing |
| DED | Directed energy deposition |
| SLM | Selective laser melting |
| SLS | Selective laser sintering |
| SEM | Scanning electron microscopy |
| DLS | Dynamic light scattering |
| PSD | Particle size distribution |
| OES | Optical emission spectroscopy |
| DIC | Digital image correlation |
| wEDM | Wire electrical discharge machining |

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
