# Peer review of "Tensile Properties of As-Built 18Ni300 Maraging Steel Produced by DED"

_applsci, doi:10.3390/app122110829_

Round 1

Reviewer 1 Report

SUMMARY

This manuscript investigated the mechanical behaviour of as-built 18Ni300 Maraging steel produced by direct energy deposition (DED) under monotonic loading. The article contains interesting comparisons of the test results of samples collected in different production directions. The article's subject area concerns the developing method of additive manufacturing of elements and may arouse interest, which justifies the article's publication. The article has a correct structure and contains well-crafted tables and illustrations, but some research results require more profound discussion. Some pictures have shortcomings that need to be corrected. Therefore, minor corrections are recommended before publication.

SUGGESTED IMPROVEMENTS

1.       Please check the title. In my opinion there is a typing error in the title, it should be DED instead of DLD.

2.      Table 3. Please provide a broader interpretation as to why the results of the research in the horizontal and vertical directions give such similar values

3.      Figures 9a, 9b, 10a and 10b . Please introduce a clearer scale.

4.      Figures 9b and 10b. Please explain such a significant difference in the size of micropores in samples taken from a material made under the same conditions, and why samples having larger micropore sizes have higher strength properties.

Figure 12. Please check the scale.

Reviewer 2 Report

In this work, authors studied the tensile behaviors of 18Ni300 Maraging steel used for DED repairing, which is a promising application in the metal AM area. The manuscript provided a good background and discussion. Here are a few comments:

1. Check some possible typos, such as in Line 93: eacceptable, in Line 230: utile.

2. In Figure 7, the caption should be "Monotonic tensile test results for the deposited bimetallic specimen".

3. In Figures 9b and 10b, it is suggested to add some arrows to indicate which is dimple and which is micropore in order to be clearer to readers.

4. In Line 219-225, more details on computing the hardening exponent should be provided. It is not very clear of the strong agreement that the comparison was made between the computed nominal ultimate tensile strength and the average true tensile strength. More descriptions can be added in this paragraph.

5. In Line 210, it seems that "The graph illustrated in Figure 12-a" should be "The graph illustrated in Figure 11".

6. In Figure 11, the hardness distribution seems to have an increasing tendency from the bottom to the top-most region, while in Figure 12, it shows that the middle (12 b) has more pores and larger pores than the near bottom region (12 a), which looks confusing. It will be better for authors to interpret this observation and if there is any correlation among hardness, porosity, and any other factors along the sample height.

Reviewer 3 Report

The study is devoted to the topical and interesting topic of restoration of parts by surfacing, performed at a good level. To improve the paper, it is good to make a few small finalizations.

Correct the Abstract a little: pay less attention to the review part and describe in general terms what has been done in the article, and then give specific values of the obtained properties.

In the Methodology section, indicate the number of manufactured samples for mechanical testing, what was their shape, and their size.

Figure 9 shows that the samples have porosity, as you write about in section 3.3. It would be good to give quantitative data on porosity for vertical and horizontal samples. It would be possible to quantify the porosity from the photographs provided by you and give the values ​​in a table or text.

When writing conclusions, it is better to remove the text on lines 234–238, it repeats the Methodology section.

You also write about three samples, but for full-fledged testing, as a rule, at least 5 samples are made per point. Why were so few samples made? This should be explained in the text of the paper.

Round 2

Reviewer 3 Report

In general, the authors made the necessary adjustments to the article and answered my questions. In my opinion the article can be published.